# Sox17 drives functional engraftment of endothelium converted from non-vascular cells

William Schachterle[1], Chaitanya R. Badwe[1], Brisa Palikuqi[1], Balvir Kunar[1], Michael Ginsberg[2], Raphael Lis[1], Masataka Yokoyama[1], Olivier Elemento[3], Joseph M. Scandura[1,4] & Shahin Rafii[1]

Transplanting vascular endothelial cells (ECs) to support metabolism and express regenerative paracrine factors is a strategy to treat vasculopathies and to promote tissue regeneration. However, transplantation strategies have been challenging to develop, because ECs are difficult to culture and little is known about how to direct them to stably integrate into vasculature. Here we show that only amniotic cells could convert to cells that maintain EC gene expression. Even so, these converted cells perform sub-optimally in transplantation studies. Constitutive Akt signalling increases expression of EC morphogenesis genes, including *Sox17*, shifts the genomic targeting of Fli1 to favour nearby Sox consensus sites and enhances the vascular function of converted cells. Enforced expression of Sox17 increases expression of morphogenesis genes and promotes integration of transplanted converted cells into injured vessels. Thus, Ets transcription factors specify non-vascular, amniotic cells to EC-like cells, whereas Sox17 expression is required to confer EC function.

[1] Division of Regenerative Medicine, Department of Medicine, Ansary Stem Cell Institute, Weill Cornell Medicine, 1300 York Avenue, Room A-863, New York, New York 10065, USA. [2] Angiocrine Bioscience, New York, New York, USA. [3] HRH Prince Alwaleed Bin Talal Bin Abdulaziz Alsaud Institute for Computational Biomedicine, Weill Cornell Medical College, New York, New York 10065, USA. [4] Department of Medicine, Hematology-Oncology, Weill Cornell Medical College and the New York Presbyterian Hospital, New York, New York 10065, USA. * These authors are co-corresponding authors. Correspondence and requests for materials should be addressed to J.M.S. (email: jms2003@med.cornell.edu) or to S.R. (email: srafii@med.cornell.edu).

For many traumatic and ischemic vascular diseases, transplanting non-lymphatic blood vessel endothelial cells (ECs) is an attractive regenerative therapy, because vascularized and functional ECs support the metabolic needs of damaged tissues[1,2] and choreograph tissue regrowth via perfusion-independent angiocrine functions[3–7]. However, assessing EC transplantation therapies is made impractical by the difficulty of purifying and expanding primary ECs that sustain their vascular fate during in vitro cultivation. A large number of stable vascular cells are necessary for testing revascularizing and regenerative EC therapies. Cellular reprogramming is appealing, because it provides a therapeutically relevant path for converting an available cell type to a transplantable EC with regenerative capabilities. Although angiogenic factors can coax non-vascular cells into EC-like lineages[8–11], the factors that drive the EC properties required to meet the translational goals of tissue regeneration have been difficult to dissociate from principal lineage specification.

To uncover factors necessary for conversion of non-vascular cells into transplantable blood vessel ECs that engraft, it is necessary to identify permissive mesenchymal or epithelial cells that are amenable to conversion into the EC identity. Pluripotent stem cells differentiate into ECs but this process is driven by pre-determined programmes that can be challenging to tease apart by reductive approaches. Moreover, ECs generated by pluripotent cells can be unstable, multipotent and/or immature[11–13]. Human amniotic fluid cells can be converted to vascular ECs (RACVECs, reprogrammed amniotic cells to vascular ECs) by overexpressing the Ets transcription factors (TFs) Etv2, Fli1 and Erg, while also inhibiting transforming growth factor-β signalling[14]. Amniotic cells, unlike pluripotent cells, are terminally differentiated, non-vascular parenchymal cells, yet they appear to retain some developmental plasticity. Amniotic cells are appealing, because they are routinely obtained from pregnant subjects with broad genetic and ethnic backgrounds[15].

Xenobiotic barriers impede thorough functional testing and direct comparison of human RACVECs to adult ECs. To facilitate in vivo functional testing of converted cells, we searched for murine cell sources that are accessible and amenable to EC conversion. Converted mouse amniotic cells (MACs), or murine RACVECs (subsequently be referred to as simply, 'RACVECs'), stably adopted an EC-like immunophenotype and acquired a transcriptome highly similar to cultured adult ECs. Despite their stable EC-like identity, murine RACVECs performed poorly in tests of EC function compared with cultured adult ECs. To identify mechanisms that might drive functional engraftment of RACVECs into host vasculature, we used constitutively active Akt signalling. Active Akt signalling is detectable in most normal adult EC beds[16] and enforced constitutive Akt signalling enables survival of cultured ES-derived and adult ECs, probably by emulating in vivo EC microenvironment cues such as tuned growth factor signals, cell–cell contacts and shear forces[17–19]. Akt signalling rescued the functional deficiencies of RACVECs by activating EC morphogenesis genes, including Sox17, and refining the Fli1 genomic binding site purview to enrich for Fli1 sites common to primary ECs including those near Sox consensus sequences. Enforced expression of Sox17 in RACVECs enabled converted cells to form stable vascular networks in vitro and in vivo, and obviated the need for constitutive Akt signalling.

Our results identify a TF network comprised of Ets and Sox17 factors that specifies and sustains EC fate and function. Although Sox17 is not required to activate a broad set of EC genes in amniotic cells, after initial EC gene induction by Ets factors, subsequent Sox17 gene regulation is required to generate long-lasting engraftable and stable ECs. As vascular engraftment after transplantation is necessary for both the angiogenic and instructive functions of ECs in orchestrating organ repair, our approach identifies a key TF network governing EC transplantation and provides an important step towards EC-directed therapy.

## Results

**Conversion of MACs to EC-like cells.** To characterize the vascular and regenerative function of reprogrammed EC-like cells, we employed well-defined congenic mouse models that overcome the confounding influence of xenografting immune-compromised mice (for example, NOD Scid Gamma) and the use of genetically disparate human cell sources. We harvested MACs from E11.5–E13.5 C57BL6/J embryos and transduced them with lentiviruses encoding mouse Ets TFs, Etv2, Erg and Fli1, and propagated the transduced cells using EC culture conditions and a transforming growth factor-β signalling inhibitor. Empty null lentivirus constructs were used as negative controls. In parallel, we attempted to convert mouse embryonic fibroblasts (MEFs) collected from E13.5 embryos and mouse adult fibroblasts (MAFs) collected from adult tail and ear tissue (Fig. 1a). Expression of the Ets TFs after transduction was similar in all three cell types as assessed by western blotting and quantitative PCR (qPCR; Supplementary Fig. 1a,b). All cell types transduced with Ets TFs lentiviruses expressed EC-linked transcripts at some point during conversion (Supplementary Fig. 1c). As the converting cells upregulated EC-associated transcripts during conversion, they also reduced non-EC genes such as the fibroblast gene Cspg4 (MAFs) and smooth muscle actin (Sma; MEFs and MACs; Supplementary Fig. 1c). We observed reduced Sma expression in Ets TF-transduced MEFs only at day 21. Hence, all three cell types tested could acquire some EC features.

Surface protein expression of VEcad and CD31 was confirmed by flow cytometry in all three cell types transduced with Ets TF-expressing lentiviruses but not in the corresponding controls (Fig. 1b and Supplementary Fig. 1d). The proportion of converted MACs expressing VEcad ($\geq 80\%$) and CD31 ($\geq 20\%$) gradually increased by day 28 (Fig. 1b). We also cultured adult mouse lung ECs (Akt-LECs) by transducing them with a constitutively active Akt signalling, to facilitate propagation ex vivo, in part, by preventing apoptosis through simulation of the microenvironment ECs experience in vivo[6,11,17–21]. Immunofluorescence (IF) staining of Akt-LECs and Ets-transduced cells showed VEcad protein loaded onto the cell surface where it coalesced at cell–cell junctions, as is characteristic of primary adult ECs (Fig. 1c). MAC-derived cells adopted the shape and morphology of Akt-LECs and co-expressed surface CD31 in a subset of converted cells (Fig. 1c). Control MAC cultures did not spontaneously convert to EC-like cells (Fig. 1b). For MAF- and MEF-derived cells, expression of both VEcad and CD31 plateaued between day 7 and day 28. By microscopy, these cells expressed low levels of VEcad and CD31 on their surface (Supplementary Fig. 1d,e). Thus, the converted MAF and MEF cells were either being outcompeted by non-converted cells or they could not maintain an EC-like phenotype during in vitro expansion.

To test whether the failure of EC-like MAF- and MEF-derived cells to stably expand was due to outgrowth by unconverted cells, we sorted transduced cells on day 7 based on their expression of CD31. We then propagated the sorted cells for an additional 4 weeks and re-analysed expression of the EC markers to examine whether the progeny retained the sorted immunophenotype (Supplementary Fig. 1f). The CD31-positive and -negative MAF fractions became indistinguishable over time. Only half of the MEFs remained CD31[+] by day 28 and a small number of CD31-neg MEFs acquired CD31 expression over time. In contrast, MAC-derived CD31[+] cells maintained their EC character with persistent expression of CD31 and VEcad, and the

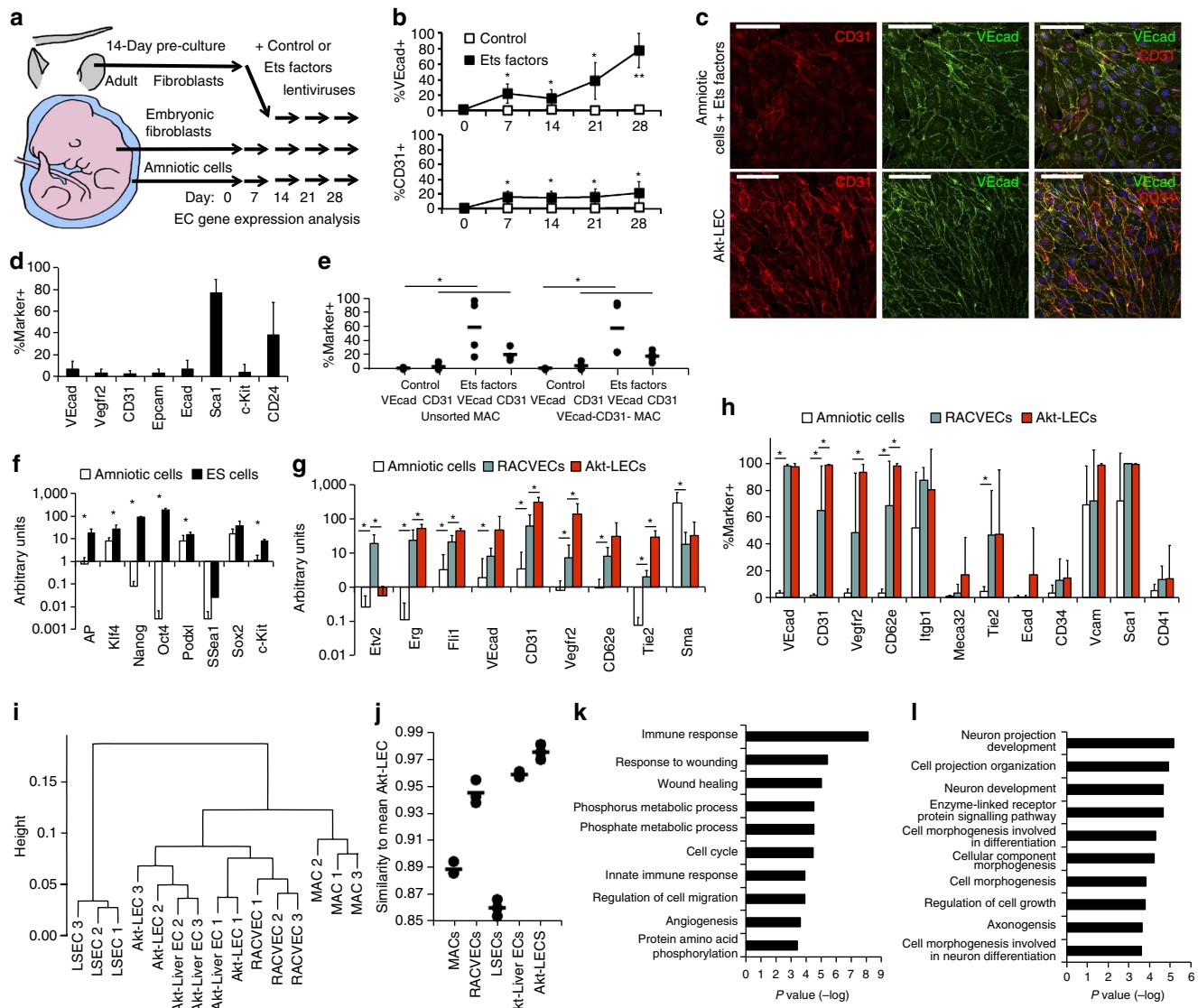

**Figure 1 | Non-vascular MACs can be converted to EC-like cells. (a)** Schematic of conversion of adult fibroblasts, embryonic fibroblasts and mid-gestation MACs. **(b)** Time course of surface expression of VEcad and CD31 on control and Ets TF-infected amniotic cells analysed flow cytometry ($n = 6$). **(c)** Surface expression of VEcad and CD31 on Ets TF–infected MACs after day 28 and on cultured Akt-LECs as detected by fluorescent microscopy. Scale bars, 100 μm **(d)** Flow cytometry indicating surface expression of EC and pluripotency markers on amniotic cells ($n = 5$). **(e)** Amniotic cells were precultured and then depleted of the VEcad$^+$ or CD31$^+$ cells and conversion efficiency was assessed by surface expression of VEcad and CD31 28 days after initiation of conversion conditions ($n = 4$). **(f)** qPCR of pluripotency factors transcript levels in amniotic cells compared to embryonic stem cells ($n = 5$). **(g)** qPCR detecting transcripts of EC markers expressed in amniotic cells, RACVECs, and Akt-LECs ($n = 7$). **(h)** Surface expression of EC markers on amniotic fluid cells, RACVECs, and Akt-LECs assessed by flow cytometry ($n = 5$). **(i)** Hierarchical clustering based on whole-transcriptome analyses of MACs, RACVECs, directly isolated liver ECs (LSECs), Akt-transduced liver ECs and Akt-transduced LECs. **(j)** An average Akt-LEC sample was calculated and 1-Pearson correlations between expression profiles were calculated to represent the proximity of each sample to the average Akt-LEC sample. **(k)** GO term analysis using the set of genes in which FPKM values were upregulated by more than log2, $P < 0.05$, for RACVECs versus amniotic cells. Genes upregulated in converted cells were enriched for EC and EC-related terms. **(l)** GO term analysis using the set of genes in which FPKM values were downregulated by more than log2, $P < 0.05$, for RACVECs versus MACs. For all panels, bar heights indicate means, error bars indicate s.d. among biological replicates, *$P < 0.05$ and **$P < 0.01$ two-sided $t$-tests, assuming normal distribution.

CD31$^-$ MAC progeny never acquired these EC-like features (Supplementary Fig. 1f). Therefore, conversion of adult and fetal fibroblasts was erratic and unstable, whereas MACs attained and stably maintained EC-like characteristics after transduction with the Ets TFs.

We wanted to ensure that the stable conversion of MACs to an EC-like cell fate was not due to the selective outgrowth of contaminating EC or pluripotent populations. MAC preparations contained small amounts of *VEcad*, *CD31* and *Vegfr2* transcripts

(Supplementary Fig. 1c), likely to be due to a small number of cells expressing endothelial markers (Fig. 1d). To exclude the possibility that MAC-derived VEcad$^+$ or CD31$^+$ cells were derived from pre-existing cells marked by EC proteins, we depleted MACs of cells expressing CD31 or VEcad and found that the non-vascular MACs could be converted just as well as the unsorted population (Fig. 1e). We could not detect significant c-kit$^+$ cells, but did find that some MACs expressed Sca1 and CD24, indicating they might express other pluripotent genes[22].

However, comparison of transcript levels of pluripotent genes in MACs to embryonic stem cells showed that MACs were largely devoid of pluripotency-associated transcripts (Fig. 1f). Therefore, these data indicate that we have directly converted non-pluripotent and non-vascular cells into EC-like cells.

Next, we comprehensively defined the molecular and functional characteristics of the converted cells. As instability of converted embryonic and adult fibroblasts precluded such analysis, we focused on the converted MACs. As mouse MACs transduced with Etv2, Erg and Fli1, and propagated in EC growth conditions for 28 days, were similar to human RACVECs in their unique EC stability, we henceforth refer to them as RACVECs. We compared the transcriptomes of MACs, RACVECs and

cultured adult mouse LECs expressing constitutively active myristoylated Akt. MACs and cultured Akt-LECs expressed much less Etv2 than RACVECs (Fig. 1g). All of the adult EC genes tested were strongly induced in RACVECs compared with MACs (Fig. 1g). In RACVECs, the expression levels of some of these genes (*VEcad* and *CD62e*) were indistinguishable from Akt-LECs, whereas others (*Erg*, *Fli1*, *CD31*, *Vegfr2* and *Tie2*) were not as highly expressed, or more variable across isolates, compared with Akt-LECs. Non-vascular specific genes, such as *Sma*, were downregulated in RACVECs to levels comparable to Akt-LECs. Transcript levels were corroborated by analysis of EC protein expression by flow cytometry (Fig. 1h and Supplementary Fig. 1g–j). Thus, murine RACVECs have stably adopted many,

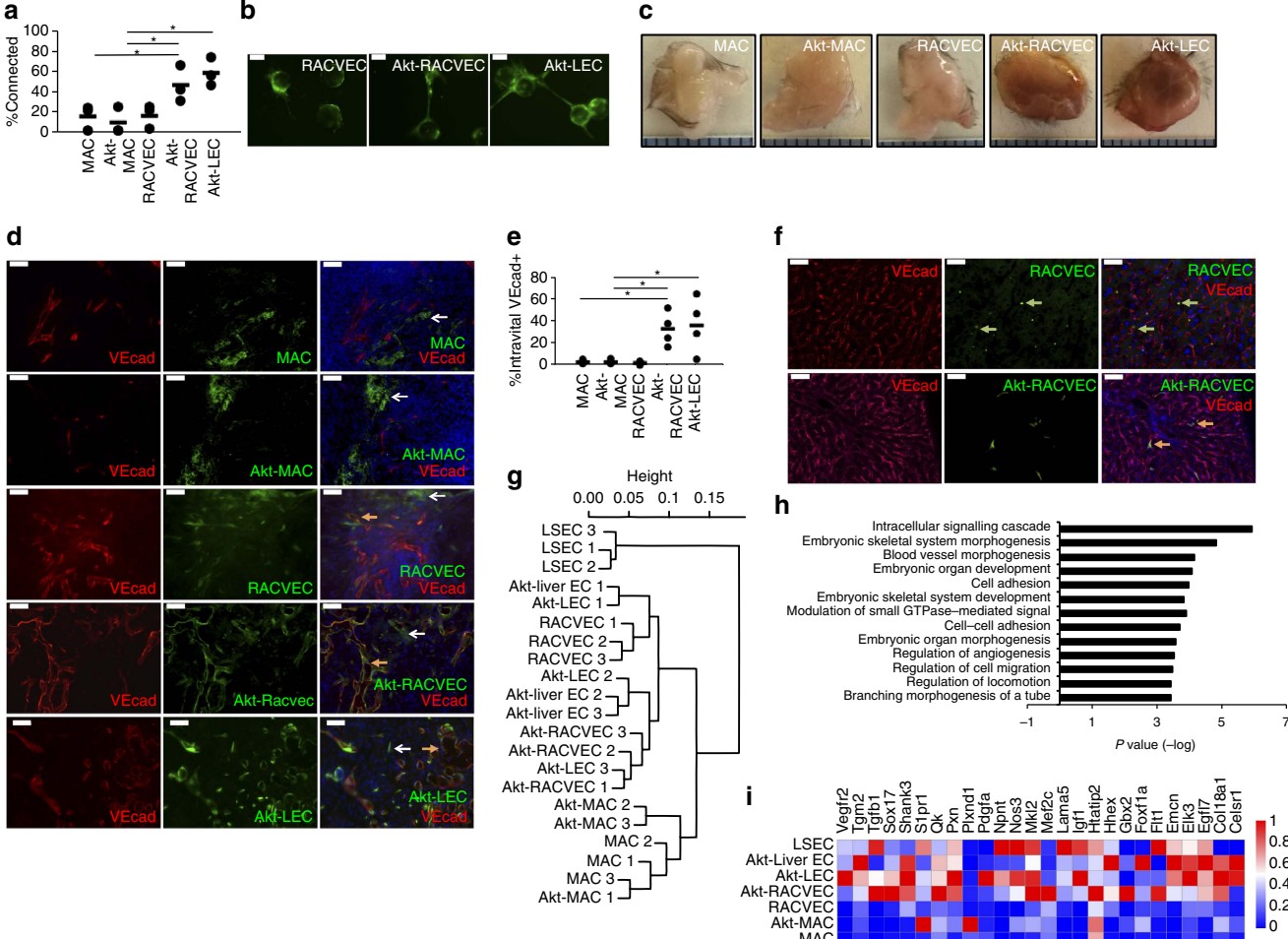

**Figure 2 | Constitutive Akt-signaling endows RACVECs with EC functions.** (**a**) *In vitro* network quantification. Branching was scored by dividing the number of beads connected to another bead by an EC tube over the total number of beads in a given field. Representative fields are shown in Supplementary Fig. 2d ($n = 3$). (**b**) EC branching, detected by fluorescent microscopy, between beads for RACVECs, Akt-RACVECs and Akt-LECs. Scale bars, 100 μm. (**c**) *In vivo* tubulogenesis of MACs, Akt-MACs. RACVECs, Akt-RACVECs and Akt-LECs. Seven days after implantation, mice were killed and plugs were dissected and photographed with a millimetre ruler. (**d**) Dissected matrigel plugs from mice that were retro-orbitally injected with fluorescently labeled anti-VEcad antibody and plugs analyzed by fluorescent microscopy. White arrows point at unincorporated GFP$^+$ cells, whereas orange arrows point at engrafted GFP$^+$ intravital VEcad$^+$ cells. Scale bars, 50 μm. (**e**) Matrigel plugs were dissociated. Resulting cell preparations were analysed by flow cytometry. The %intravital VEcad$^+$ on the *y* axis represents the percent of labeled cells recovered from the plug that were also stained by intravital fluorescently labelled anti-VEcad antibody ($n = 4$). (**f**) Mice were hepatectomized and injected intrasplenically with fluorescently labelled RACVECs or Akt-RACVECs. Fourteen days later they were retro-orbitally injected with fluorescently labelled anti-VEcad antibody and analysed by fluorescent microscopy. Green arrows point at GFP$^+$ cells, orange arrows point at engrafted GFP$^+$ intravital VEcad$^+$ cells. Scale bars, 50 μm. Three independently isolated Akt-RACVECs were tested in three experiments and for all isolates, we observed engraftment. (**g**) Hierarchical clustering based on whole-transcriptome analyses. (**h**) GO term analysis using the set of genes in which FPKM values were increased by more than log2, $P < 0.05$, for Akt-RACVECs versus RACVECs. (**i**) Heatmap with genes from the vessel and branching morphogenesis GO categories shown in Fig. 2h. Colours reflect *Z*-scores of individual isolates with blue representing 0 and red representing the maximum FPKM value, 1, for a given transcript. For all panels, bar heights indicate means, error bars indicate s.d. among biological replicates, *$P < 0.05$ and **$P < 0.01$ in two-sided *t*-tests, assuming normal distribution.

but not all, morphologic, transcriptional and immunophenotypic features of ECs.

We sequenced the messenger RNA of MACs, RACVECs and cultured ECs, and as an additional control we included non-lymphatic ECs directly purified from mouse livers (Supplementary Fig. 1k), referred to as LSEC, which were a readily available abundant source of non-cultured ECs. We found that RACVECs clustered more closely to cultured ECs than they did to MACs (Fig. 1i). We extracted the distances from the clustering analysis and observed that RACVECS were more similar to cultured primary LECs than MACs were (Fig. 1j). Differences in gene expression between cultured and directly purified cells were paramount. This separation was not apparent when we restricted the clustering to genes associated with the Gene Ontology (GO) term 'angiogenesis', suggesting that our cultured cells maintained the essential characteristics of ECs and the differences between cultured and directly purified cells were attributable to their disparate environments (Supplementary Fig. 1l). As our goal was to analyse cells that can be propagated in vitro for eventual transplantation, we focused our analysis on differences among the cultured cell types. On average, 624 genes were upregulated in RACVECs compared with MACs and pathway analysis indicated that this group was enriched for EC-associated functions and angiogenesis (Fig. 1k). Moreover, 674 genes were downregulated in RACVECs compared with MACs and, for this group, the most enriched pathways were neuron projection development and morphogenesis involved in differentiation (Fig. 1l). We performed genomic sequencing of cultured MACs, RACVECs and directly purified LSECs, and, similar to human amniotic cells and human RACVECs, identified no gross genetic alterations in the cultured cells compared with the directly purified adult ECs, indicating genomic stability (Supplementary Fig. 1m). Hence, RACVECs broadly and thoroughly adopted an EC-like identity and conversion erased the original MAC identity.

**Akt signalling enhances RACVEC endothelial function**. Our ultimate goal is to generate abundant, stable vascular cells that can form new vessels after transplantation and engraft into existing vessels. However, clear interpretation of human RACVEC functions in vivo was unavoidably obstructed by donor cell heterogeneity and the technical requirement for xenograft analysis. Accordingly, we tested the ability of RACVECs to form vascular networks in vitro using a fibrin bead assay. RACVECs formed as few connections as MACs and less than Akt-LECs (Fig. 2a,b). Thus, it appeared that despite the phenotypic and molecular similarity of RACVECs to primary ECs, the converted cells were missing key programmes necessary for vascular functions.

Constitutively active Akt signalling enables stable propagation of functional ECs; thus, we tested whether this approach could confer vascular functions to RACVECs. RACVECs transduced with a lentivirus driving expression of myristoylated Akt (Akt-RACVECs) could be stably propagated in vitro, as assessed by morphological and immunophenotypic analyses (Supplementary Fig. 2a–c). Akt signalling did not alter the transcript levels of the Ets factors (Supplementary Fig. 2d). Akt-RACVECs and RACVECS did not express high levels of genes associated with tumour EC phenotypes compared with control cultured ECs (Supplementary Table 1)[23].

To determine whether Akt signalling improved the EC function of RACVECs, we used in vitro surrogates and direct in vivo tests. Unlike RACVECs, Akt-RACVECs readily formed connections in vitro (Fig. 2a,b and Supplementary Fig. 2e). We injected matrigel plugs loaded with green fluorescent protein (GFP)-marked MACs, Akt-MACs, RACVECs, Akt-RACVECs or

Akt-LECs subcutaneously into the flanks of C57Bl/6 mice. After 7 days, we retro-orbitally injected fluorescently labelled anti-VEcad antibody to visualize functional, perfused vessels (intravital VEcad$^+$). On removal of the plugs, we found that only the matrigel grafts with Akt-RACVECs and Akt-LECs were perfused with blood, as indicated by reddish colouring of recovered material (Fig. 2c). Examination of the plugs by fluorescent microscopy showed that Akt-RACVECs and Akt-LECs readily engrafted into vessels anastomosed with the recipient vasculature (GFP$^+$ intravital VEcad$^+$), whereas the vast majority of RACVECs were unincorporated (GFP$^+$ intravital VEcad$^-$) into host vessels (Fig. 2d). We digested the plugs to quantify the percentage of injected cells integrated into the recipient circulatory system. RACVECs displayed minimal incorporation (appearing as GFP$^+$ intravital VEcad$^-$), similar to negative controls MAC and Akt-MAC. In contrast, Akt-RACVECs and Akt-LECs incorporated into and anastomosed with the host vasculature (GFP$^+$ intravital VEcad$^+$; Fig. 2e). We also used a model of organ regeneration in which 70% partial hepatectomy triggers compensatory liver regrowth requiring neo-angiogenesis and angiocrine support[5,24]. Although both Akt-RACVECs and RACVECs could be identified in the regenerating liver, only Akt-RACVECs were incorporated into vasculature of regenerating livers as reported by intravital VEcad staining (Fig. 2f). Hence, Akt-RACVECs, but not RACVECs, performed morphogenic functions necessary for EC tube formation and engraftment into recipient blood vessels.

**Akt signalling modulates RACVEC gene expression**. To assess how constitutively active Akt signalling functionalizes RACVECs, we compared the transcriptomes of MACs, RACVECs, Akt-MACs, Akt-RACVECs, Akt-LECs and murine liver ECs (Akt-Liver ECs), and directly purified liver ECs. Constitutively active Akt signalling did not make the global transcriptional profile of MACs or RACVECs more similar to adult primary ECs, as assessed by K-means distance and principle component analysis (Fig. 2g and Supplementary Fig. 2f,g). The relative similarity of Akt-RACVECs to Akt-LECs or Akt-Liver ECs was no greater when the analysis was restricted to genes within the 'angiogenesis' GO and directly isolated ECs clustered with cultured EC and converted cells (Supplementary Fig. 2h). Analysis of endothelial TFs and EC markers showed that some genes were differentially expressed, but there was no obvious pattern (Supplementary Fig. 2i). We focused on genes differentially expressed in the cultured cells, because our goal was to identify factors contributing to in vitro stable propagation and eventual transplantation. Within the set of genes more highly expressed in Akt-RACVECs than RACVECs were genes associated with adhesion, and vessel and tube morphogenesis, consistent with the functional deficiencies observed in RACVECs (Fig. 2h). Pathways enriched in the set of Akt-downregulated genes revealed that adhesion genes were also enriched, reinforcing the notion that Akt signalling alters the ability of converted cells to stably connect with extracellular matrix (ECM) and other cells (Supplementary Fig. 2j). The relative expression level of differentially regulated genes within the morphogenesis-associated ontologies is shown in Fig. 2i. Among these genes upregulated were EC TFs, specifically *Mef2c*, *Tal1*, *Elk3*, *Hhex* and *Sox17*, as well as matrix and receptor-signalling proteins *Col18a1*, *Emcn*, *Shank3*, *Vegfr2* and *CD31*. Thus, Akt-RACVECs are broadly similar to RACVECs, but activate EC genes that confer EC tubulogenic and morphogenic functions.

**Akt signalling modifies the Fli1 genomic binding in RACVECs**. In contrast to ES-derived and adult ECs, Akt signalling was not

required for stable RACVEC culture and, as such, RACVECs could be used to parse the endothelial functions of Akt signalling separate from its pro-survival effects *in vitro*. We speculated that Akt signalling might support *in vivo* function by modifying the genomic Fli1 binding site purview to enrich EC gene targeting and extinguish binding to non-vascular genes.

We used chromatin immunoprecipitation (ChIP) sequencing to directly test this hypothesis by comparing the Fli1 genomic binding purviews in RACVECs, Akt-RACVECs, Akt-LECs and freshly isolated liver ECs. With this approach we identified ~11,000 of Fli1 differentially bound regions (DBRs) that were shared by all cell types, probably representing a 'Core' endothelial signature (Fig. 3a and Supplementary Fig. 3a–c). These core DBRs typically occurred near 5′-regulatory regions of annotated refGene transcripts (Fig. 3b, left). We used data from the ENCODE project[25] to map EC regulatory regions identified in human umbilical vein EC (HUVEC) to the mouse genome and found that Fli1 DBRs occurred near EC promoter and enhancer regions (Fig. 3b, right), including genomic sites near *VEcad*, *CD31* and *Vegfr2*, genes expressed in ECs and activated during the conversion process (Fig. 3c). Thus, the Fli1 binding purviews in converted cells is highly enriched for regions involved in vascular gene regulation.

Although clustering and PCA analyses of transcriptional profiles indicated that Akt signalling did not broadly enhance EC gene expression, we found that Akt signalling refined the Fli1 genomic binding site purview of RACVECs, making it more similar to Akt-LECs and LSECs (Fig. 3d and Supplementary Fig. 3d). Whereas Akt-RACVECs shared ~90% of their Fli1-binding sites with Akt-LECs and ~64% with LSECs, only about 67% of the DBRs in RACVECs were shared with Akt-LECs and ~35% with LSECs (Supplementary Fig. 3b). The increased similarity with freshly isolated LSECs indicated that the genomic purview change observed was not simply a result of myr-Akt, but reflective of the ability of Akt signalling to simulate the *in vivo* environment.

To uncover how constitutive Akt signalling increased the similarity of the Fli1 purview in RACVECs, we generated data sets composed of sites differentially bound by Fli1 in Akt-RACVECs compared with RACVECs. These sets of DBRs were called 'Akt-RACVEC Up' and 'Akt-RACVEC Down' (Fig. 3e). We further restricted the Akt-RACVEC Down set and created a 'RACVEC Unique' set composed of ~7,000 sites that were downmodulated by Akt signalling in RACVEC (that is, Akt-RACVEC down) and also not enriched in either Akt-LEC or LSEC (Fig. 3e and Supplementary Fig. 3e). DBRs belonging to the Akt-RACVEC Up set were bound by Akt-LECs and LSECs, supporting the notion that Akt signalling mimics the native EC environment (Fig. 3e and Supplementary Fig. 3f). We generated three derived chromatin state maps: genomic regions marked with a defining chromatin state in any cell type (Composite), regions marked by a chromatin state in HUVEC but not other cell types (HUVEC Unique) and regions marked by a chromatin state in cell types other than HUVEC (HUVEC Excluded). We found that the Core EC, RACVEC unique and Akt-RACVEC Down sites were associated with promoter and enhancer regions found in all of the cell types analysed (Fig. 3f)[26]. RACVEC Unique and Akt-RACVEC Down sites were enriched at genomic regions that were marked as promoter and enhancer elements in non-HUVEC (Fig. 3f). In contrast, sites belonging to Akt-RACVEC Up were associated with regions uniquely marked as promoters or enhancers in HUVEC (Fig. 3f, HUVEC Unique) and relatively excluded from regulatory regions absent in HUVEC (Fig. 3f, HUVEC Excluded and other cell types). We also analysed the overlap of sites in these sets with the human Fli1 purview in HUVECs in Patel *et al.*[27] and found that the Core and

Akt-RACVEC Up sets overlapped with murine regions that corresponded to regions identified as human Fli1 targets far more likely to be than by chance (Fig. 3g). Therefore, Akt signalling extinguishes Fli1 binding to extraneous binding sites and drives binding to genomic sites key to EC identity and function.

To identify the TFs that might bind to key EC regulatory regions, we performed *de novo* motif discovery using the genomic sequences of Fli1 DBRs identified in each cell type and data set, and searched for known motifs in the JASPAR database (Supplementary Tables 2–7)[28]. As expected, a canonical GGAA Ets binding motif was found within all Fli1 DBRs and the most highly preferred Ets-like motif, (A/C)GGAA(G/A), was very similar in all cell types and DBR sets studied. CCAAT box motifs were preferentially enriched with Fli1 DBRs in RACVECs and in the RACVEC unique set, but not in the other cell types or in the set of Core Fli1 DBRs (Fig. 3h). In contrast, a Sox consensus motif was strongly enriched in Fli1 DBRs found in Akt-LECs and Akt-RACVECs but not in DBRs found in RACVEC Unique. Furthermore, an unbiased search for accessory motifs near the most highly enriched Ets motif identified Ebox and Sox sites in the Akt-RACVEC Up set (Supplementary Fig. 3g). Finally, Sox sites were enriched in the sequences in the regions differentially bound by Fli1 in the Akt-RACVECs versus RACVECs (Supplementary Table 7). Thus, Akt signalling modifies Fli1 genomic binding site selection, favouring regions unique to ECs that often contained Sox motifs.

As multiple Sox family members have been implicated in EC development and function[29], we searched for a specific Sox factor that might be associated with the Sox binding motif. The relevant Sox factor should be expressed in Akt-RACVECs and Akt-LECs but not in RACVECs. We found that only *Sox17* met these criteria, suggesting that Akt signalling both induced *Sox17* expression and repositioned Fli1 genomic binding towards Ets sites with nearby Sox motifs (Fig. 3i). Western blotting confirmed that Sox17 protein was present in Akt-RACVECs, Akt-LECs and LSECs, but absent in RACVECs (Supplementary Fig. 3h). These results raised the possibility that Sox17 could replace Akt signalling to enhance EC morphogenic function in RACVECs and promote engraftment.

**Sox17 augments RACVEC EC function**. We hypothesized that enabling Sox17 expression in RACVECs could replace constitutively active Akt signalling and rescue the defects of RACVECs. To test this, we enforced expression of Sox17 in RACVECs and tested whether this could rescue RACVEC defects using the fibrin and matrigel plug vascularization assays (Supplementary Fig. 4a). We found that enforced Sox17 expression in RACVECs (Sox17-RACVECs) enhanced *in vitro* EC vascular function (Fig. 4a,b and Supplementary Fig. 4b). Sox17-RACVECs engrafted into lumenized, anasto-mosed vessels perfused with blood in matrigel plugs (Fig. 4c–e). Similarly, Sox17-RACVECs engrafted into lumenized intravital VEcad$^+$ blood vessels when transplanted into mice after partial hepatectomy (Fig. 4f).

To assess engraftment and function of Sox17-RACVEC in an assay of revascularization, we injected GFP-marked RACVECs, Akt-RACVECs, Sox17-RACVECs or saline (control) intra-muscularly after partial femoral artery excision. We intravitally labelled cells exposed to circulation by injecting labeled anti-VEcad antibody and found that at day14 Akt-RACVECs and Sox17-RACVECs were incorporated into host vasculature (Fig. 4g). We corroborated our microscopy results using flow cytometry of digested day 14 thigh muscle tissue (Fig. 4h). Although transplanted Akt-RACVECs and Sox17-RACVECs displayed engraftment, only Sox17-RACVECs

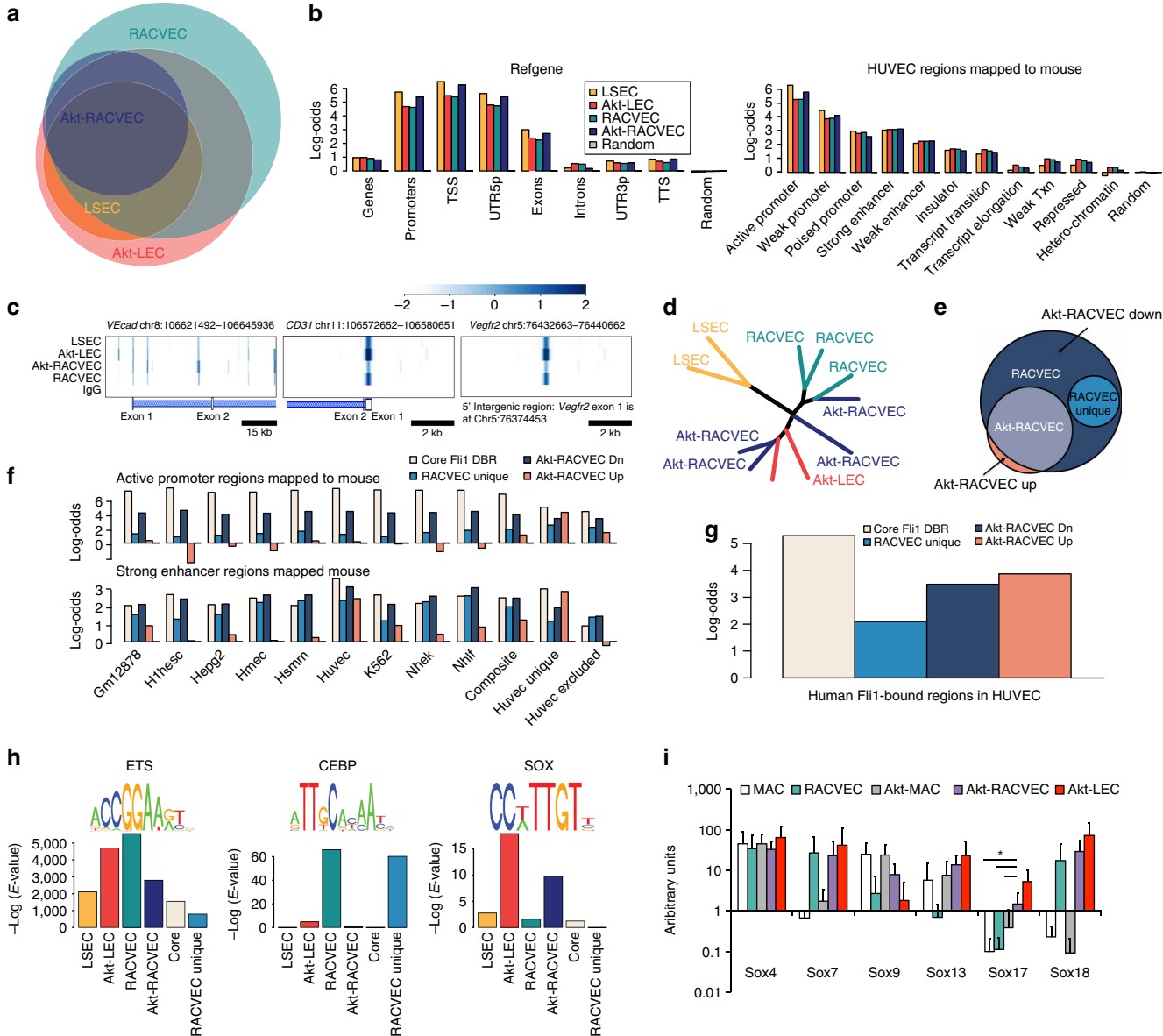

**Figure 3 | Akt signalling steers the genomic targeting of Fli1 towards endothelial genes.** (**a**) Venn diagram indicating overlapping DBRs. (**b**) The log-odds ratio of the overlap between the indicated refGene elements (top graph) and the genomic locations of the indicated Fli1 DBRs is shown. Grey is a control set of random genomic locations with the same median bp length as the Fli1-bound regions. The lower graph indicates the overlap between the indicated regulatory regions derived from ENCODE chromatin states for HUVECs and the genomic locations of the Fli1-bound regions for each cell type. (**c**) Heatmaps of selected loci of EC genes. The mean binding signal for biological replicates is shown in blue scale. Gene structures are represented with white bars (exons) and blue (introns). (**d**) Dendrogram of Fli1-binding in all samples depicts similarity by *k*-means distance/clustering based on normalized Fli1-binding logFC. (**e**) Venn diagram indicating overlapping and unique regions. Purple represents sites in Akt-RACVEC set and the darker blue represents RACVEC sites that are downmodulated compared with Akt-RACVEC, the light blue represents sites that are absent in all other cell types but RACVEC and the slice of orange represents sites upmodulated in Akt-RACVECs. (**f**) Overlap between murine elements homologous to chromatin states ENCODE defined as active promoter and the indicated Fli1-binding site sets. The probabilities are indicated on the *y* axis and the cell types whose regulatory regions were analysed are indicated on the *x* axis. (**g**) Overlap between murine genomic elements homologous to Fli1-bound regions in HUVECs[27] and the genomic locations of Fli1-bound regions. (**h**) Enrichment of motifs representative of specific families is shown for the indicated datasets. Inverse log *E*-values are indicated on the y-axis (higher is more enriched). (**i**) qPCR detecting transcripts of members of the Sox family of TFs among MAC, Akt-MAC, RACVEC, Akt-RACVEC and Akt-LEC ($n = 5$). Bar heights indicate means, error bars indicate s.d. among biological replicates tested in experiments performed at different times, *$P < 0.05$ according to two-sided *t*-tests, assuming normal distribution.

enhanced reperfusion by days 14 and 21 compared with control (Fig. 4i and Supplementary Fig. 4c,d). We were able to observe unincorporated RACVECs at day 1 after surgery (Supplementary Fig. 4e), indicating that their absence at later times might be due to their inability to stably incorporate into vessels. Indeed, failure to recover RACVECs at later stages could be due to passive

dispersal of cells or cell death as a consequence of their failure to establish homotypic and/or ECM interactions[30]. By contrast, Sox17-RACVEC and Akt-RACVEC engraftment was long-lasting and vessel-integrated cells could be observed two months after transplantation (Supplementary Fig. 4f). Hence, Sox17 functionalizes transcriptionally converted EC-like cells so they

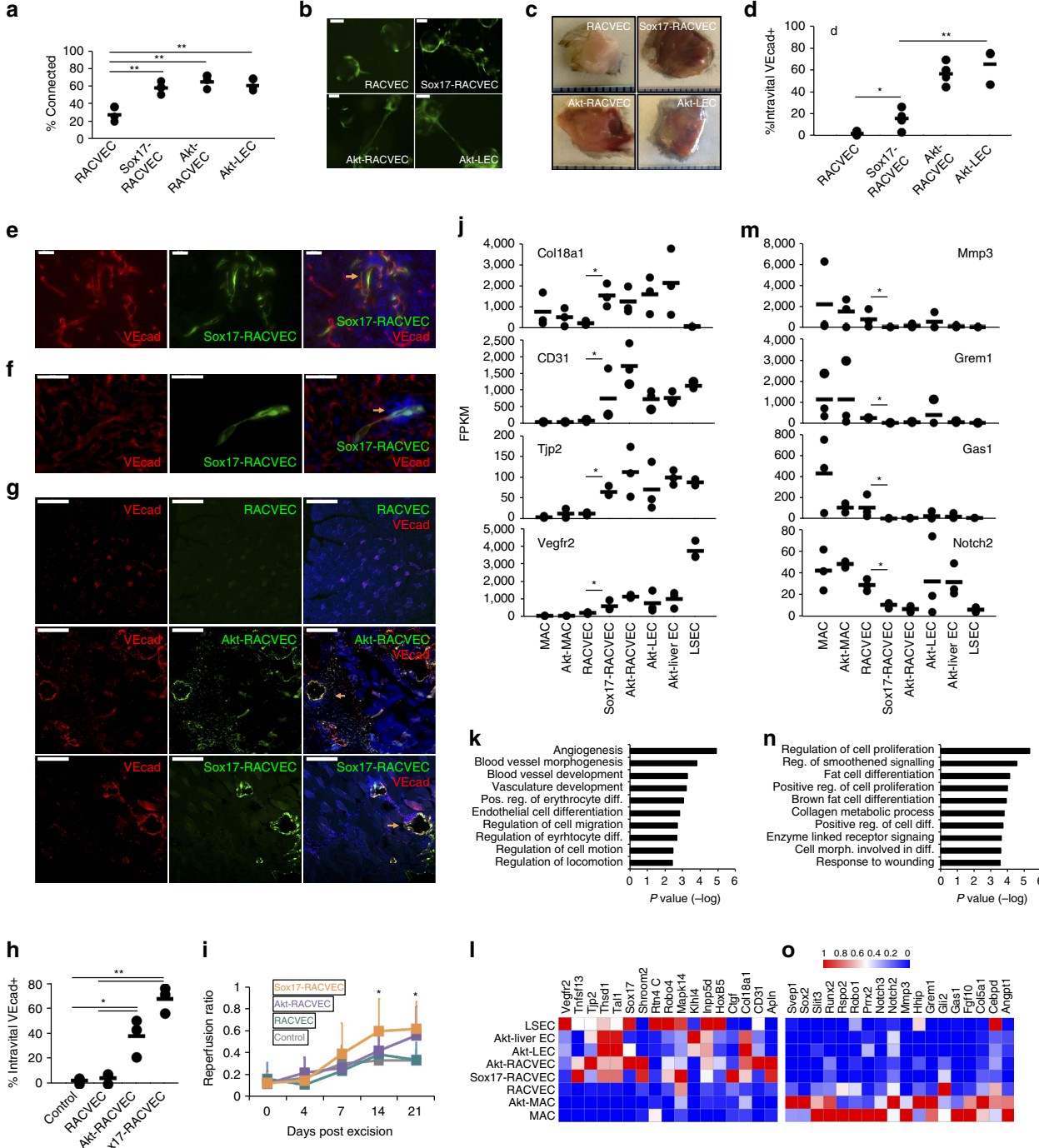

**Figure 4 | Sox17 enhances conversion and endows RACVECs with EC functions.** (**a**) *In vitro* network quantification. Representative fields are shown in Supplementary Fig. 4b ($n = 3$). (**b**) EC connections between beads with Sox17-RACVECs. Scale bar, 100 μm. (**c**) *In vivo* tubulogenesis of RACVECs, Sox17-RACVECs and Akt-LECs. Plugs were dissected and photographed with a millimeter ruler. (**d**) Matrigel plugs were fixed and sectioned and analyzed by fluorescent microscopy. Orange arrows point at engrafted GFP[+] intravital VEcad[+] cells. Scale bars, 50 μm ($n = 3$). (**e**) Matrigel plugs were dissociated and analyzed by flow cytometry. The %intravital VEcad[+] represents the percent of labelled cells recovered from the plug that were stained by intravital anti-VEcad antibody ($n = 3$). (**f**) Two mice were hepatectomized and injected intrasplenically with GFP-labelled Sox17-RACVECs. Liver sections were analysed by fluorescent microscopy. Orange arrows point at engrafted GFP[+] intravital VEcad[+] cells. Scale bars, 50 μm. (**g**) Mice that underwent unilateral artery excision and cell transplantation were injected with fluorescently labelled anti-VEcad antibody and killed. Images of mice from day 14. Orange arrows point at engrafted GFP[+] intravital VEcad[+] cells. Scale bars, 200 μm. (**h**) Thigh muscles were dissociated and analysed by flow cytometry 14 days after injury and cell injection. The %intravital VEcad[+] represents the percent of labeled cells recovered from the muscle that were stained by intravital anti-VEcad antibody ($n = 3$). (**i**) At days 1, 4, 7, 14 and 21 reperfusion was measured as the ratio of intensity measured in the affected limb over the unaffected limb ($n = 8$). Bar heights indicate means, error bars indicate s.d. among individual male animals, *$P < 0.05$ and **$P < 0.01$ two-sided $t$-tests assuming normal distribution. (**j**,**m**) FPKMs indicating transcript levels of indicated genes ($n = 3$). (**k**,**n**) GO term analysis using the set of genes in which FPKM values were increased by more than log2, $P < 0.05$, for Sox17-RACVECs versus RACVECs and vice versa. (**l**,**o**) Heatmaps of representative cell types with selected genes. Colours reflect $Z$-scores of individual isolates with blue representing 0 and red representing the maximum FPKM value, 1.

can perform as stable *bona fide* blood vessel ECs after transplantation.

We compared the transcriptomes of Sox17-overexpressing RACVECs with our other EC populations. We found that 141 genes were more highly expressed in Sox17-RACVECs than RACVECs and 30% of these genes were also found in the constitutive Akt-induced set. The set included *Col18a1*, *CD31*, *Tjp2 (ZO-2)* and *Vegfr2* (Fig. 4j). Similar to the genes induced by Akt signalling, the Sox17-induced group was enriched for morphogenesis-associated genes (Fig. 4k). The relative expression of selected morphogenesis- and angiogenesis-associated genes are shown in Fig. 4l. We found that 337 genes were downregulated and 30% of these genes were also found in the set of genes that were reduced in Akt-RACVECs. The most strongly reduced genes included *Mmp3*, *Grem1*, *Gas1* and *Notch2* (Fig. 4m). Although the genes downregulated by Akt signalling were associated with cellular interactions and metabolic processes (Supplementary Fig. 2j), those downregulated by Sox17 were associated with proliferation and smoothened signalling (Fig. 4n). Genes associated with the most significantly enriched pathways are shown in Fig. 4o. Thus, Sox17 modifies a group of genes that overlaps with those affected by constitutive Akt signalling and is enriched with genes associated with morphogenesis, cellular interactions and proliferation.

## Discussion

Engineered ECs that can be expanded, transplanted and engraft into compromised blood vessels to restore their perfusion and instructive functions could significantly alter how vascular diseases are studied, treated and cured[1,2]. As ECs play essential roles in organ development and regeneration, engineered engraftable ECs could help repair or replace injured tissue[3–6] or even reverse disease[31–33]. Although specific human ECs, such as HUVECs, can be propagated in culture, murine ECs tend to drift to non-vascular cells and cease proliferating. For this reason, our approach of direct conversion of mouse mid-gestation amniotic cells to ECs compares favourably with previous reports of EC production from fibroblasts and pluripotent sources because of the stability and efficiency of RACVEC generation. Direct conversion of MAFs to ECs indicated that, similar to our data, adult fibroblast-derived ECs are unstable and do not maintain their EC identity over time[34]. Human fibroblasts[35] and adipose tissue[36] can be converted to EC-like cells by Etv2 and Sox18, respectively, but xenobiotic barriers limited analysis of the converted cells to *in vitro* assays and short-term tests in immunocompromised mice. Thus, vascular durability *in vivo* could not be assessed in these pioneering studies. Many groups, including our own, have explored the potential of generating ECs from human and mouse pluripotent cells or employing pluripotency factors[8–11,13]. However, these valuable studies highlight the issue that initially led us to evaluate amniotic cells as potential source cell type for lineage conversion studies: the differentiated ECs derived from pluripotent cells are often unstable and drift into non-vascular lineages. RACVEC lineage stability may be a consequence of a gene or set of genes expressed, or not expressed, in amniotic cells, compared with fibroblasts. Alternatively, RACVEC stability may be attributable to a favourable chromatin state that allows TF access to a restricted set of stabilizing genes but not to genes that lead to fate instability/drift.

Transcriptome analysis of basic Ets-converted RACVECs indicated that they activated a broad range of genes expressed in cultured ECs and were more similar to cultured ECs than to their starting amniotic cell identity. RACVECs performed EC functions poorly, suggesting that endothelial functions require expression of genes not induced after enforced overexpression of Ets TFs and/or dynamic cellular adaptations that were not supported in the converted cells. We took advantage of the finding that microenvironment cues activate Akt to identify Sox17 as a factor that promotes engraftment, despite its being dispensable for broad EC gene activation.

The genomic targeting of Fli1 in converted cells was broadly similar to Fli1-binding in cultured and freshly purified ECs. Deeper analysis uncovered a novel role for constitutive Akt signalling to refine the Fli1 genomic purview, shedding non-EC genomic targets in favour of a more minimal EC purview. Fli1 sites shed in the presence of Akt signalling were enriched for CCAAT-box binding proteins (CEBP family members) that are more closely associated with haematopoietic and adipocyte activities. As Fli1 is essential for haematopoietic function, enforced expression of Fli1 may drive it to bind promiscuously to many accessible Ets motifs, regardless of whether they regulate EC or non-EC genes. It is plausible that Akt signalling shifts Fli1 binding either by altering affinity of Fli1 for particular Ets motifs or by shifting the repertoire of accessory factors with which Fli1 can bind.

Akt signalling activated Sox17 expression and enriched Fli1 binding sites with nearby Sox motifs. Sox17 is more highly expressed by arterial cells that experience high shear[30] and by tip cells[37]. The fact that we did not see members of the Notch family associated with arteriogenesis upregulated by Sox17 could be a result of differences between the conversion process and EC differentiation or instability of arteriovenous identity in cultured ECs[38]. Nevertheless, our data show that Sox17 activates a subset of genes induced by Akt signalling that includes genes associated with morphogenesis, and cellular and ECM interactions. A simple model based on the data is that Sox17 activates genes that stabilize barrier and basal matrix interactions, while downregulating destabilizers.

Therapeutic replacement of blood vessels has been difficult to establish, because so little is known about how ECs might weave themselves into existing vascular infrastructure. Transplanted ECs can improve recovery after injuries and disease and multiple studies in mouse models have shown improved reperfusion after ischemic limb injuries[9,34,35]. However, these findings have not been borne out in clinical studies, as intramuscular delivery of bone marrow-derived cells to patients with peripheral artery disease is ineffective[39]. This discordancy could be due to the differences in injury-type, species or source cells. In addition, it is unclear whether engraftment and neovascularization, as we and others have observed in mouse models, improve clinical measures of vascular and muscular health.

This work defines a mouse-based tractable lineage conversion strategy for engineered ECs and identifies a novel regulator of EC repair functions that could be used to enhance therapeutic EC incorporation into injured vessels. This approach sets forth a platform with which the mechanisms that underlie key EC angiogenic and instructive functions can be reductively tested and identified and translated to the clinical setting.

## Methods

**Cell culture.** Mid-gestation amniotic cells were isolated from pregnant females that were wild type or tdTomato reporter mice in C57BL/6 background, timed according to date (E0.5) of the observed vaginal plug. Amniotic sacs containing embryos aged E11.5–E13.5 were removed from the maternal uterus and placed in saline (PBS). The yolk sac and amnion were gently disrupted, exposing the embryo and releasing the amniotic fluid. The amniotic fluid was washed with PBS and cultured in AmnioMAX (Invitrogen 12001-027), AmnioMAX supplement (Invitrogen 12556-023) and 1 × Pen/Strep (Invitrogen 15240-062). After 5 days, the unattached cells were removed by changing media. The cells were cultured for a total of 14 days at 37° in 5% $CO_2$ before conversion was initiated. Typically, one litter, containing 5–10 embryos, would yield 0.5–1 × $10^6$ amniotic cells.

MEFs were isolated from E13.5 C57BL/6 embryos. Fetal livers and heads were removed, and the remaining tissue was minced with a sterile razor blade. Preparations were incubated with 0.05% trypsin/EDTA (LDP 25-052-CI) for 30 min at 37° in 5% $CO_2$, further disrupted by pipetting up and down, washed and then plated in DMEM (LDP 15-013-CM) containing 10% fetal bovine serum (FBS), $1 \times$ Pen/Strep (Invitrogen #15240-062). MEFs were pre-cultured for 14 days.

MAFs were isolated from C57BL/6 adult tail tip and ear tissue, which were minced with a sterile razor blade and incubated with 0.05% trypsin/EDTA (LDP 25-052-CI) for 30 min at 37° in 5% $CO_2$, then further disrupted by pipetting up and down. Trypsin was quenched by adding equal volume of FBS and the mixture was diluted in DMEM (LDP 15-013-CM) containing 10% FBS, $1 \times$ Pen/Strep (Invitrogen 15240-062). MAFs were pre-cultured for 14 days. We considered a biological replicate as an independently isolated culture of MACs or MEFs, which consisted of mixed gendered embryos, and MAFs, which consisted of fibroblasts taken from one adult female.

Cells undergoing conversion were grown in EC media, which was composed of 1:1 low glucose DMEM:F12 (LDP 10-013-CV, LDP 10-080-CV), 20% FBS, $1 \times$ Pen/Strep (Invitrogen 15240-062), $1 \times$ non-essential amino acids (LDP 25-060-Cl), 10 mM HEPES (Invitrogen 15630-080), 100 µg ml$^{-1}$ heparin (Sigma-Aldrich H3149), 50 µg ml$^{-1}$ endothelial mitogen (Alfa Aeser J65416) and 5 µM SB431542 (R&D 1614) on tissue culture plastic coated with fibronectin (Sigma F1141). During the 28-day conversion process, EC media was supplemented with 20 ng ml$^{-1}$ mouse VEGF-A (Peprotech 450-32).

For Akt-LEC or Akt-Liver ECs, lung or liver ECs were isolated by magnetic cell sorting, using sheep anti-rat IgG Dynabeads (Life Technologies) pre-captured with an anti-CD31 antibody. Cells were plated and allowed to grow for one day in EC media, then infected with a lentivirus encoding constitutively active myrAkt1 (Akt). One to 3 weeks later, cells were re-purified by FACS using anti-VEcad and anti-CD31 antibodies. Directly purified Liver ECs were isolated by FACS after mice were injected intravitally with fluorescently labeled anti-VEcad and Isolectin[20]. For ChIP experiments with LSECs, approximately 6 adult mice and $1.5 \times 10^7$ of purified cells were used per isolate.

**Lentiviral vectors and transduction.** Mouse Etv2, Erg, Fli1 and Akt complementary DNAs were cloned into the pCCL-PGK lentivirus vector. Mouse Sox17 cDNA was cloned into Lv203 (Genecopeia) lentivirus vector. Viruses were produced in 293T cells, concentrated with Lenti-X concentrator (Clontech 631232) and titred using the Lenti-X p24 Rapid Titer kit (Clontech 632200). Cells were infected using MOI 10. Sox17-RACVECs were produced by infecting RACVECs with lentivirus bearing Sox17 and performing puromycin selection.

**Antibodies.** The antibodies used were as follows:
Anti-mouse Etv2 (Abcam, EPR5229(2); western blotting 1:100)
Anti-mouse Erg (Abcam, 9FY; western blotting 1:100)
Anti-mouse Fli1 (Abcam, ab15289; western blotting 1:100)
Anti-mouse CD31 (Biolegend clone 390; flow cytometry 1:2,000)
Anti-mouse VE-Cadherin (Biolegend clone BV13; flow cytometry 1:500)
Anti-mouse Vegfr2 (DC101; flow cytometry 1:500)
Anti-mouse CD62e (BD Biosceinces, clone 10E9.6; flow cytometry 1:500)
Anti-mouse Itgb1 (BD Biosciences, clone 18/CD29; flow cytometry 1:500)
Anti-mouse Meca32 (BD Biosciences, clone MECA-32; flow cytometry 1:500)
Anti-mouse Tie2 (Biolegend, clone Tek4; flow cytometry 1:500)
Anti-mouse ECadherin (BD Biosciences, Clone 36/E-Cadherin; flow cytometry 1:500)
Anti-mouse CD34 (BD Biosciences, Clone RAM34; flow cytometry 1:500)
Anti-mouse Vcam (BD Biosciences, Clone 429 MVCAM.A; flow cytometry 1:500)
Anti-mouse Sca1 (Biolegend, clone D7; flow cytometry 1:500)
Anti-mouse CD41 (Biolegend, clone MWReg30; flow cytometry 1:500)
Anti-mouse CD24 (BD Biosciences clone M1/69; flow cytometry 1:500)
Anti-mouse VE-Cadherin (R + D AF1002; IF 1:100)
Anti-mouse CD31 (Biocare, Clone Mec13.3; IF 1:100)
Uncropped western blotting images are shown in Supplementary Fig. 5.

**RNA analysis.** RNA was prepared using RNAeasy Mini kit (Qiagen 74106) and 1 µg was converted to cDNA using qScript cDNA SuperMix (Quanta 95048-100). Relative transcript levels were determined by qPCR, performed on a 7500 Fast Real Time PCR System (Applied Biosystems) using SYBR Green PCR Master Mix (Applied Biosystems). No RT or template control and inspection of dissociation curves verified amplifications. Arbitrary units were determined by normalizing to Gapdh levels. Primer sequences are shown in Supplementary Methods.

RNA was prepared similarly for RNA Sequencing and the quality was checked on an Agilent Technologies 2100 Bioanalyzer. Libraries were prepared using the TruSeq RNA sample Preparation Kit (Illumina Rs-122–2001) and sequenced as $2 \times 51$ bp reads at the Weill Cornell Genomics Core Facility with the Illumina HiSeq2000 sequencer using paired-end module. After quality control using the Illumina pipeline, reads were mapped using Tophat with default parameters and mouse genome build mmp9 (ref. 40). Cufflinks with upper-quartile normalization

and sequence-specific bias correction was used to generate Fragments per kilobase of transcript per million fragments (FPKM) values[41].

Hierarchical clustering and principal component analysis were performed in R using log2 transformed FPKM values with distances calculated by subtracting the Pearson's correlation value from 1. Those values were also used to generate the bar graphs indicating distances to an average cultured lung EC sample. Clustering was unsupervised except when only genes associated with the GO term 'angiogenesis' were used. Pathway analysis was performed using the set of differentially expressed genes for an indicated comparison. Genes were considered differentially expressed if their log2 fold change was greater than or less than 1 for the comparison of their averaged FPKMs and if the $P$-value was $< 0.05$ according to a two-sided $t$-test, and not assuming equal variance. Terms among the top 10 GOTERM_BP_FAT category, as determined by DAVID, were used[42]. Heatmaps were generated using the pheatmap function in R after normalizing FPKM values by the maximum value for a given transcript.

**Genomic stability.** DNA from fresh/cultured cells was isolated and purified using PureLink Genomic DNA Mini Kit (Invitrogen K182001). DNA library preps were prepared and multiplexed and used as input for low coverage whole-genome sequencing with HiSeq 100, producing 100 bp single-end reads. Sequencing reads were de-multiplexed (bcl2fastq), checked for quality (FastQC) and trimmed/filtered when appropriate (Trimmomatic). The resultant high-quality reads were mapped (BWA-MEM) to the UCSC mm9 genome build. Uniquely mapped reads were then binned (BEDTools) into 10 K bp-sized stepwise, non-overlapping window tiles spanning the mm9 genome build sequence. The resultant read counts data was then used to generate segmentation profiles for chromosomes 1–19, X and Y.

Counts were normalized to the total number of mapped reads per sample and scaled with the total number of reads in the reference sample. To ensure real value ratios, the scaled reads were adjusted by $+1$ count. Log2 ratios were acquired by performing log2-transformation of the adjusted and scaled experimental count over the adjusted and scaled reference count. To identify genomic regions with abnormal copy number, we implemented a circular binary segmentation algorithm[43] on our DNA copy number data to partition genomic regions that were divergent. We used the R-package DNAcopy to plot segmentation profiles (via circular binary segmentation algorithm) for each sample in reference to LSECs. Default parameters were employed for all data processing commands.

**Fibrin bead assay.** Method was adapted for use with cultured mouse cells from Nakamatsu et al.[44]. Cells ($5 \times 10^5$) were incubated with 1260 Cytodex 3 Collagen beads (GE Life Sciences 17-0485-01) overnight at 37° in 5% $CO_2$ in EC media. The next day, the cells/beads were washed and resuspended in 2 mg ml$^{-1}$ fibrinogen (Sigma-Aldrich F8630) in PBS with 0.15 U ml$^{-1}$ aprotonin (Roche 10236624001). This mix was plated with 1.25 U ml$^{-1}$ thrombin (Sigma 82050-844) and allowed to form a solid matrix. Cell/bead/matrix preparations were cultured in StemPro-34 Serum-Free Media (Invitrogen 10639-011), reconstituted according to the manufacturer's instructions and supplemented with $1 \times$ L-glutamine (Thomas Scientific #B003L18), $1 \times$ Pen/Strep (Invitrogen 15240-062), $1 \times$ β-mercaptoethanol, 0.5 mM Ascorbic Acid (Millipore), 200 µg ml$^{-1}$ bovine holotransferrin (Sigma Aldrich T1283), 100 µg ml$^{-1}$ heparin (Sigma-Aldrich H3149), 20 ng ml$^{-1}$ mouse VEGF-A (Peprotech 450-32) and 20 ng ml$^{-1}$ FGF2 (Peprotech 100-18B). After 4 days, EC connections were scored by counting the number of matrix-suspended beads connected by cells and dividing by the total number of beads in a given field. Each experiment included three wells of cell/bead mixes and images were taken of each well and the mean connected percentage across those wells were averaged to generate one n. Additional controls, which contained no thrombin, were used to confirm comparable cell survival across cell types and experiments.

***In vivo* engraftment by Matrigel and partial hepatectomy.** Cells were transduced with mCherry or GFP-expressing lentivirus and purified by FACS. Cells ($2 \times 10^6$) were suspended in PBS containing 100 µg ml$^{-1}$ heparin (Sigma-Aldrich H3149), 20 ng ml$^{-1}$ mouse VEGF-A (Peprotech 450-32) and 20 ng ml$^{-1}$ FGF2 (Peprotech 100-18B), mixed 1:1 with Matrigel (BD Biosciences 354234) and injected subcutaneously into the flanks of C57BL/6J mice. Seven days later, mice were retro-orbitally injected with fluorophore-labelled anti-VEcad antibody and killed. Plugs were removed, photographed and, if being analysed by microscopy, fixed in 4% paraformaldehyde then embedded in OCT. If incorporation was being quantitated, plugs were enzymatically digested with 2.5 mg ml$^{-1}$ Collagenase A (Roche 11088793001) and 1 unit ml$^{-1}$ Dispase II (Roche 04942078001) for 30 min at 37 °C under gentle agitation, filtered and analyzed by flow cytometry.

For hepatectomies, the right medial, left medial and left lateral lobes of C57Bl/6 mice were resected with silk suture (Roboz) after anaesthetization with 100 mg kg$^{-1}$ ketamine and 10 mg kg$^{-1}$ xylazine. The sutures were used to tie off the individual lobes, one at a time, and scissors were used to but the lobe just distal to the suture to minimize injury and blood loss. GFP- or tdTomato-labelled cells ($5 \times 10^5$) were resuspended in PBS and injected intrasplenically into mice that underwent 70% partial hepatectomy, as described previously[20]. After 14 days, mice were retro-orbitally injected with fluorescently-labelled anti-VEcad, killed and the organs were fixed, mounted and prepared for imaging by fluorescent microscopy.

**ChIP analysis.** Anti-Fli1 or control Rabbit IgG (Ab46540) were used to identify Fli1-bound regions using a method based on detailed protocol report[20]. Cells ($2-5 \times 10^7$) were fixed in 1% paraformaldehyde diluted in EC media. Fixation was quenched with 125 mM glycine and the cells were washed three times with PBS. After nuclei isolation and sonication using a Bioruptor, chromatin-protein complexes were incubated with 10 μg antibody bound to Dynabeads M-280 (Invitrogen) overnight at 4 °C under gentle agitation. Complexes were washed with PBS containing 0.5% BSA and 5 mM EDTA using magnetic separation and then DNA purified by phenol-chloroform extraction. Enrichment was tested by qPCR, paired-end (75/75 bp) libraries were produced and then sequenced at the MSKCC Integrated Genomics Operation on Illumina HiSeq 4000. Sequences were mapped to the mm9 genome reference using bwa mapper. Analysis of Fli1 DBRs was done using a combination of SAMtools and custom R and Python scripts to carry out general linear modeling[45]. Salient features of this analytical framework are the use of a negative binomial error model and appropriate false discovery rate corrections to identify Fli1 DBRs associated with a particular phenotype. This method controls the type 1 error while preserving good detection power for differential binding. Contrast models were used to identify differential Fli1 sites in the different cell types. DBRs were prioritized by the fold change between groups (Fli1 v IgG or between cell types) and the corrected P-value, to identify the genomic elements with the most robust group effect on the Fli1 binding purview. Motif analyses were performed with tools from the MEME suite[28].

**Hindlimb ischaemia.** The proximal part of the femoral artery and the bifurcation point between the popliteal artery and the saphenous artery of C57Bl/6 mice were ligated and all side braches were dissected. The femoral artery was excised out via ligated points. With the muscle still exposed, $5 \times 10^5$ cells suspended in PBS were injected into the gracilis muscle. Hindlimb reperfusion was measured at indicated time points with laser Doppler perfusion imager (Persican PM3, Perimed). For quantification or visualization, one randomly selected member of each group was selected and, retrorbitally injected with fluorescently-labelled anti-VEcad antibody and sacrificed to visualize tissue. Fixed and mounted tissue sections were analysed by fluorescent microscopy.

**Statistics.** Significance of pairwise comparisons were determined using unpaired Student's t-tests with a significance threshold at $P < 0.05$. All values are presented as mean with error bars, indicating s.d. For genomic stability, a cutoff threshold of $3+$ absolute s.d. was used for segment split calling. For ChIP sequencing, selection criteria were set at absolute value (log fold change) $> 1.5$, false discovery rate $< 0.01$, logCPM $> -3$.

**Code availability.** For specific requests, please contact the corresponding author.

**Data availability.** The data that support the findings of this study are presented within the manuscript and Supplementary Files. Sequencing data have been deposited in the GEO repository accession codes GSE85642 http://www.ncbi.nlm.nih.gov/geo/query/acc.cgi?acc=GSE85642.

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

## Acknowledgements

We thank Dr Michael Poulos, Dr Jason Butler and Michael Gutkin for helpful discussions and experimental assistance. We acknowledge Dr Jenny Xiang and the CLC Genomics and Epigenomics Core Facility at Weill Cornell, as well as Dr Agnes Viale and the Integrated Genomic Operation at Memorial Sloan Kettering Cancer Center for assistance with high-throughput sequencing. W.S. was supported by the NIH training grant T32HL94284 and a NYSTEM training grant (NYS C026878). M.Y. was supported by Manpei Suzuki Diabetes Foundation, Mochida Memorial Foundation for Medical and Pharmaceutical Research, and the Uehara Memorial Foundation. J.M.S was supported by NIH-NHLBI (HL1166436, HL119872, HL128158), Taub Foundation, Cancer Research and Treatment Fund (CR&T), Leukemia & Lymphoma Society (2299-14), Tri-Institutional Stem Cell Initiative (TRI-SCI #2014-023 and #2016-024), the Empire State Stem Cell Board and the New York Sate Department of Health grants (C029156, C030160, ECRIP-Empire Clinical Research Investigator Program). S.R. was supported by Ansary Stem Cell Institute (ASCI), Empire State Stem Cell Board and New York State Department of Health (C0: 26878, 28117), HHMI, NHLBI (R01s: HL115128, HL119872 and HL128158), the NIDDK (R01DK095039) and the Qatar National Priorities Research Program (6-131-3-268).

## Author contributions

W.S., J.M.S. and S.R. conceived and designed the study. W.S., C.B., B.P., M.G., R.L. and M.Y. performed experiments. W.S., B.K., O.E. and J.M.S. analysed data. W.S. wrote the manuscript. S.R. and J.M.S. oversaw the project; all authors discussed the results and commented on the manuscript.

## Additional information

**Competing Financial Interests:** M.G. is a senior scientist at Angiocrine Bioscience. The remaining other authors declare no competing financial interests.

