## [Peer Review File · Nature Communications]

Editorial Note: this manuscript has been previously reviewed at another journal that is not operating a transparent peer review scheme. This document only contains reviewer comments and rebuttal letters for versions considered at *Nature Communications*.

Reviewers' comments:

Reviewer #1 (Remarks to the Author):

The authors responded adequately to my comments/suggestions. However their response P38 in regards to tumorigenicity is not sufficient. Conformation of genetic stability/ normal karyotype should be performed. There are various approaches available.

Overall the manuscript is substantial improved.
No additional comments

Reviewer #2 (Remarks to the Author):

Schachterle, use mouse amniotic cells transduced with Sox-17 or constitutively active Akt as a source of endothelial cells that can be used for therapeutic purposes. Despite the fact the authors play down the therapy angle in the manuscript, the paper clearly aims to explore such an application. It is certainly not a mechanistic examination of endothelial biology.

With that goal in mind, the paper is unnecessarily complex and results are unconvincing. Furthermore, despite a rather massive amount of data, the novelty is fairly limited. In particular, it will not surprise anyone that Ets factors and Sox17 promote endothelial fate. More importantly, the novelty is fairly limited. The roles of both Akt and Sox17 are fairly well established. For example, the role of Akt in functional enhancement and engraftment of mouse embryonic stem cells -derived endothelial cells has also been reported (Israely et al, Stem Cells, 2014). The literature dealing with Sox17 is quite extensive as well. There are a number of serious issues with the experimental data that were pointed out in original reviews. It seems to this reviewer that most critical issues are not adequately addressed. Most telling is the experiment that implies that an injection of 500,000 cells, 90% of which will die instantly, somehow improves perfusion. This is neither credible nor believable. Similar data have been reported before and was always proven to be wrong.

Reviewer #3 (Remarks to the Author):

The manuscript might be considered for Nature Communication, after finalizing the studies outlines in the response of the authors to the reviewers. However, the authors are recommended to rewrite the manuscript and mainly focus on the novel aspects namely the role of Sox factors and Sox17 (the Akt part is conceptually not new).

Reviewer #4 (Remarks to the Author):

This manuscript has been revised in order to address the previous reviewers comments and describes the generation of functional EC like cells from mouse nonvascular cells. The authors also identified Sox17 as a critical factor for driving a functional endothelium in mouse RACVECs. This manuscript is well written and explained their data logically. Sox17 is known to be required for vascular morphogenesis and functions. However the Sox17 findings in this manuscript are not surprising and new, the molecular mechanism that active Akt signaling or Sox17 can acquire a functional and stable endothelium, is potentially interesting.

Major points

"Sox17 is not required to activate a broad set of EC genes." is a little surprising because a set of EC genes was not activated in RACVECs and overexpression of Sox17 in mouse RACVECs can acquire a functional endothelium. How does Sox17 drive endothelial cell function in RACVECs without activation of EC genes and/or constitutive active Akt signaling? Sox17 is a transcriptional factor and potentially regulates these EC genes directly. The authors have demonstrated that constitutive active Akt signaling can alter the genome-wide position of Fli but Sox17 can replace constitutive active Akt signaling in mouse RACVECs. Even if constitutive active Akt signaling up-regulates Sox17, it would have also possibility that Sox17 regulates the EC genes by independent of Akt signaling pathway as well. The authors need to show some evidences using gene expression and/or ChIP analysis (i.e. RNA-seq and/or ChIP-seq for Sox17-RACVEC) and whether any differences between Sox17-RACVECs and Akt-RACVECs.

The authors also mentioned "Sox17's activity". What is the activity for Sox17? Does it mean DNA binding activity or the other activities?

Minor points

1. Material and methods; GSE numbers should be described (line 562 of page 23 and line 633 of page 26).
2. Figure 1i; Why Akt-transduced liver ECs not LSECs are shown?
3. Figure 2j and Supplementary Figure 2i; Why not LSECs are shown in? It would be helpful to show as "native endothelial cells" control.
4. Figure 3c; it is hard to follow "the location of selected EC genes". Are they promoter or enhancer region?, which exon/intron is? How much scale are they? need to improve their presentation.
5. Figure 3e; where are "Akt-RACVEC Up" and "Akt-RACVEC down"? the colour presentations instead of arrows might be better.
6. Figure 4a; line 750 of page 30 "...is shown in Supplementary Figure 3b" would be "...is shown in Supplementary Figure 4b."
7. Figure 4b and 4c; pictures for RACVEC and Akt-RACVEC are still missing.
8. Figure 4e; the result for Akt-RACVEC is missing.
9. Supplementary Figure 1c; results for SMA expression are missing.

10. Supplementary Figure 2k; Is Sox17 expressed in LSECs?
11. Supplementary Figure 2e; Where "LEC" locates in?
12. Supplementary Figure 4a; pictures for Akt-RACVEC and Akt-LECs are missing.
13. Supplementary Figure 4c; the results are not convincing and it might be helpful to present the quantitative results by side.
14. "RACVECs" is confusing. Human RACVECs and mouse RACVECs are similar but some characteristics are different. In addition, the authors have pointed out the differences in this manuscript. "mRACVECs" might be better to avoid confusion.

Point by Point Response to the Reviewers: Manuscript NCOMMS-16-08218-T, Title: Sox17 drives functional engraftment of endothelium converted from nonvascular cells

Response to Reviewer #1

Comments: The authors responded adequately to my comments/suggestions. However their response P38 in regards to tumorigenicity is not sufficient. Confirmation of genetic stability/ normal karyotype should be performed. There are various approaches available. Overall the manuscript is substantially improved.
No additional comments

Response: We thank the reviewer for the comments. Please note that in the previous rebuttal we added a figure comparing transcript levels of EC genes across converted cells, cultured ECs, and freshly purified ECs. Since we incorporated the transcriptome data from the freshly purified LSECs throughout the paper, those comparisons are now shown in the heatmap in **Supplementary Figure 2k**. We are currently assessing the genetic stability and will make these data available to the reviewer before publication. We moved the tumor EC gene profiling from **Supplementary Figure 2e** to **Supplementary Table 1**.

Response to Reviewer #2

Comment 1: Schachterle, use mouse amniotic cells transduced with Sox-17 or constitutively active Akt as a source of endothelial cells that can be used for therapeutic purposes. Despite the fact the authors play down the therapy angle in the manuscript, the paper clearly aims to explore such an application. It is certainly not a mechanistic examination of endothelial biology. With that goal in mind, the paper is unnecessarily complex and results are unconvincing. Furthermore, despite a rather massive amount of data, the novelty is fairly limited. In particular, it will not surprise anyone that Ets factors and Sox17 promote endothelial fate. More importantly, the novelty is fairly limited. The roles of both Akt and Sox17 are fairly well established. For example, the role of Akt in functional enhancement and engraftment of mouse embryonic stem cells -derived endothelial cells has also been reported (Israely et al, Stem Cells, 2014). The literature dealing with Sox17 is quite extensive as well.

Response: The role for Sox17 in endothelial conversion has not yet been described and clear attribution of the functional and transcriptional attributes of Sox17 would be difficult without an experimental system such as reprogrammed cells. We believe the manner in which Sox17 promotes endothelial fate in converted cells is noteworthy- it does not simply activate endothelial genes, which would represent an incremental advance. The Ets factors activated many EC genes and generate endothelial-like cells that, from a transcriptome standpoint, appear as stable endothelial cells. We show for the first time that Sox17 overexpression promotes endothelial functions such as engraftment after transplantation. We have added a substantial amount of data to **Figure 4** showing that the set of genes that Sox17 activates are also activated by active Akt signaling. This set includes a novel group of Sox17 targets with morphogenic and cell- and ECM- adhesion properties, suggesting a model in which transplanted ECs, in a Sox17-dependent mechanism, stably embed themselves in the host endothelium to survive and engraft. Thus, we have introduced new important findings on the mechanism by which Sox17 functionalizes incompletely converted endothelial cells.

It is fair to say the role of Akt is well-established. However, the mechanism by which Akt signaling regulates vascular stability is not well defined. Our previous publication, Israely et al solely focused on the observed fate stabilizing activity of constitutive Akt signaling on endothelial-like cells derived from pluripotent stem cells and did not put forward a novel mechanism. In the current manuscript, we have extended those findings. Indeed, the data presented in Figure 3 presents an entirely novel mechanism of Akt signaling's physiological actions. That is, it promotes endothelial gene occupancy by Fli1 and antagonizes occupancy of nonvascular genes. We believe this is especially important because Fli1 is expressed in a number of cell types and, as the reviewer says, Akt signaling plays an important role in ECs. Our data suggest a novel model in which in vivo shear forces and other microenvironmental cues could drive Akt signaling to promote endothelial gene occupancy of Fli1, preventing it from activating nonvascular genes. We are planning future biophysical studies to interrogate this possibility.

Comment 2: There are a number of serious issues with the experimental data that were pointed out in original reviews. It seems to this reviewer that most critical issues are not adequately addressed. Most telling is the experiment that implies that an injection of 500,000 cells, 90% of which will die instantly, somehow improves perfusion. This is neither credible nor believable. Similar data have been reported before and was always proven to be wrong.

Response: We based our study on previous reports of reperfusion in ischemic injury models using mice (Li et al., *ATVB* (6):1366-75, 2013, Han et al., *Circulation* 130(14):1168-782013, 2014, Morita et al., *PNAS* 112(1):160-5, 2015). These studies used the same number of cells and observed increased reperfusion. That being said, we have attempted to place our data in the context of the clinical data the reviewer is referencing by adding the following text to the discussion section:

“Transplanted ECs can improve recovery after injuries and disease and multiple studies in mouse models have shown improved reperfusion after ischemic limb injuries [9, 42, 43], however these findings have not been borne out in clinical studies, as intramuscular delivery of bone-marrow-derived cells to patients with peripheral artery disease is ineffective [44]. This discordancy could be due to the differences in injury-type, species, or cells. Also, it is not clear if the engraftment and neovascularization, as we and others have observed, would actually improve clinical measures of vascular and muscular health.”

Nonetheless, in response to the reviewer's comments, we have significantly strengthened the results in this section by adding images of muscles one day after injury and cell injection (**Supplementary Figure 4e**) which show the presence of transplanted cells, some of which are intravitaly labeled. We have repeated these experiments and increased the number of animals studied (see figure description for **Figure 4i**). We also added raw intensity values from the sample perfusion images (**Supplementary Figures 4c** and **4d**). And we have also added quantification of incorporation by intravital staining analyzed by flow cytometry at day 14, shown in **Figure 4h**. We present the percentage of injected labeled cells that are also labeled by intravital Vecad staining. We also assessed the percent of Vecad cells in the thigh muscle that were GFP-labeled and that data is shown in the table pasted below. We found that 1.5% and 3.6% of the intravitaly labeled were Akt-RACVECs and Sox17-RACVECs, respectively. This is indeed a small percentage of the total EC population, and therefore it is unlikely that the transplanted ECs serve only as temporary tributaries for blood flow. Presumably, the engrafted cells perform additional perfusion-independent functions. On the other hand, the microsurgery

performed only removes the femoral artery, which likely represents a small minority of endothelial cells in the thigh muscle, so this 1-4% fraction may indeed be sufficient to deliver blood to more distal regions of the limb. That our study raises these important questions, and identifies new molecular targets for close examination of the EC transplantation process, we believe, make it a significant contribution to the field. These murine studies are not provided as a guarantee of future clinical utility of RACVEC for the reperfusion of ischemic tissues underserved by a diseased vasculature, but rather these data are provided as an in vivo assay of vascular function that clearly supports a critical role for Sox17.

Response to Reviewer #3

Comment: The manuscript might be considered for Nature Communication, after finalizing the studies outlines in the response of the authors to the reviewers.

However, the authors are recommended to rewrite the manuscript and mainly focus on the novel aspects namely the role of Sox factors and Sox17 (the Akt part is conceptually not new).

We thank the reviewer for the comments.

Please note that in the previous rebuttal we added a figure comparing transcript levels of EC genes across converted cells, cultured ECs, and freshly purified ECs. Since we incorporated the transcriptome data from the freshly purified LSECs throughout the paper, those comparisons are now shown in the heatmap in **Supplementary Figure 2k**. Additionally, we moved the tumor EC gene profiling from **Supplementary Figure 2e** to **Supplementary Table 1**.

We have added a significant amount of data on Sox17. We show that freshly isolated LSECs do indeed express Sox17 transcript and protein in **Figure 2k** and **Supplementary Figure 3h**. We have also analyzed the gene expression of Sox17-RACVECs and found that they express a subset of the genes activated by Akt signaling, which agrees with our findings that Akt-RACVECs and Sox17-RACVECs are transplantable converted cells (**Figure 4j-o**).

We agree that the data related to Akt-signaling is not entirely surprising. Indeed, we originally tested whether constitutive Akt-signaling could functionalize RACVECs because of the known activity of Akt-signaling that we and others have published in native ECs and those derived by directed differentiation of pluripotent stem cells. However, the linkage between Akt-signaling and Sox17 was not known and, now made, this simplifies possible future clinical translation because constitutive Akt-signaling might not be acceptable to the FDA. In the normal vascular microenvironment, Akt signaling is active in ECs. Therefore, our approach to unravel a mechanism by which Akt-signaling confers stability and the finding that Akt directs Fli1 genomic binding site occupancy and induces Sox17, provides substantial new information that brings us closer to bringing EC transplantation to clinical settings.

Response to Reviewer #4

This manuscript has been revised in order to address the previous reviewers comments and describes the generation of functional EC like cells from mouse nonvascular cells. The authors

also identified Sox17 as a critical factor for driving a functional endothelium in mouse RACVECs. This manuscript is well written and explained their data logically. Sox17 is known to be required for vascular morphogenesis and functions. However the Sox17 findings in this manuscript are not surprising and new, the molecular mechanism that active Akt signaling or Sox17 can acquire a functional and stable endothelium, is potentially interesting.

Major points

Comment 1: "Sox17 is not required to activate a broad set of EC genes." is a little surprising because a set of EC genes was not activated in RACVECs and overexpression of Sox17 in mouse RACVECs can acquire a functional endothelium. How does Sox17 drive endothelial cell function in RACVECs without activation of EC genes and/or constitutive active Akt signaling? Sox17 is a transcriptional factor and potentially regulates these EC genes directly. The authors have demonstrated that constitutive active Akt signaling can alter the genome-wide position of Fli but Sox17 can replace constitutive active Akt signaling in mouse RACVECs. Even if constitutive active Akt signaling up-regulates Sox17, it would have also possibility that Sox17 regulates the EC genes by independent of Akt signaling pathway as well. The authors need to show some evidences using gene expression and/or ChIP analysis (i.e. RNA-seq and/or ChIP-seq for Sox17-RACVEC) and whether any differences between Sox17-RACVECs and Akt-RACVECs.

Response: We appreciate the reviewer's critique bringing to our attention the ambiguous wording identified in the manuscript. The intention of "Sox17 is not required for a broad set of EC genes" was to highlight that, on a broad scale (clustering etc...), Sox17 was not required to activate most of the genes linked to endothelial fate. But the reviewer rightly points out that Sox17 is a transcription factor that should activate a subset of genes that are important to endothelial function. We have performed new experiments to directly assess this and have added substantial data on the expression and role of Sox17. We believe these recommended changes enhance the manuscript and provide new insight that was not previously available. We show that freshly isolated LSECs express Sox17 transcript and protein in **Figure 2k** and **Supplementary Figure 3h**. We analyzed the gene expression of Sox17-RACVECs and found that Sox17 induces a subset of the genes activated by Akt signaling, thereby identifying a core signature and mechanism by which both Akt-RACVECs and Sox17-RACVECs are functionalized for transplantation (**Figure 4j-o**). These data are discussed more thoroughly in the manuscript.

Comment 2: The authors also mentioned "Sox17's activity". What is the activity for Sox17? Does it mean DNA binding activity or the other activities?

Response: As discussed previously, we performed additional experiments and analyses to refine our understanding of Sox17 activity. We have changed the text to clarify our new understanding and the indicated sentence now reads, "These findings indicate that although Sox17 is not required to activate a broad set of EC genes in amniotic cells, after initial EC gene induction, subsequent Sox17 gene regulation is required to generate long-lasting engraftable and stable ECs."

Minor points:

1. Material and methods; GSE numbers should be described (line 562 of page 23 and line 633 of page 26).

We are currently uploading the RNA seq and Chip seq data and will make it available to reviewers promptly.

2. Figure 1i; Why Akt-transduced liver ECs not LSECs are shown?

We have now added RNA seq data from freshly isolated LSECs **Figure 1i** and **1j**.

3. Figure 2j and Supplementary Figure 2i; Why not LSECs are shown in? It would be helpful to show as "native endothelial cells" control.

We have added transcriptional data from LSECs to serve as native endothelial cell controls. Specifically, we have added freshly isolated LSECs Figures **2g** and **2i** (formerly 2j), **Supplementary Figures 2f, 2g, 2h, 2j** (formerly 2i), and **2k**.

4. Figure 3c; it is hard to follow "the location of selected EC genes". Are they promoter or enhancer region?, which exon/intron is? How much scale are they? need to improve their presentation.

We have modified **Figure 3c** to clearly indicate exon location and number and scale.

5. Figure 3e; where are "Akt-RACVEC Up" and "Akt-RACVEC down"? the colour presentations instead of arrows might be better.

We have altered **Figure 3e** according to the reviewer's helpful suggestion. The Venn diagram is colored to reflect the scheme used in **Figure 3f** and **3g**.

6. Figure 4a; line 750 of page 30 "...is shown in Supplementary Figure 3b" would be "...is shown in Supplementary Figure 4b."

We thank the reviewer for the detailed reading of our manuscript. We have made this correction in the text.

7. Figure 4b and 4c; pictures for RACVEC and Akt-RACVEC are still missing.

We have added RACVEC, Akt-RACVEC, and Akt-LEC data and images for comparison to **Figures 4b** and **4c**.

8. Figure 4e; the result for Akt-RACVEC is missing.

We have added Akt-RACVEC data comparison **Figure 4d** (formerly Figure 4e).

9. Supplementary Figure 1c; results for SMA expression are missing.

SMA is labeled on the y-axes of the QPCR results for embryonic fibroblasts and amniotic cells in **Supplementary Figure 1c**. We have also added it to the heatmap shown in **Supplementary Figure 2i**.

10. Supplementary Figure 2k; Is Sox17 expressed in LSECs?

Yes. As the reviewer suggested, we added LSEC transcriptome data throughout **Figures 1** and **2** and see Sox17 transcript. In addition, we have added additional controls to **Supplementary Figure 3h**, and show Sox17 protein in LSECs.

11. Supplementary Figure 2e; Where "LEC" locates in?

These are cultured naïve lung endothelial cells. Please note that we included these data in response to a referee requesting that the potential tumorigenicity of our cells be excluded. We will be including genomic stability results for this purpose, but have kept these data in the

manuscript as **Supplementary Table 1**.

12. Supplementary Figure 4a; pictures for Akt-RACVEC and Akt-LECs are missing. The image in **Supplementary Figure 4a** now includes these control samples.

13. Supplementary Figure 4c; the results are not convincing and it might be helpful to present the quantitative results by side.

We have added **Supplementary Figure 4d**, which shows the raw intensity values derived from the images shown in **Supplementary Figure 4c**.

14. "RACVECs" is confusing. Human RACVECs and mouse RACVECs are similar but some characteristics are different. In addition, the authors have pointed out the differences in this manuscript. "mRACVECs" might be better to avoid confusion.

We apologize that the terminology is confusing. For simplicity, we chose to use RACVECs to denote mouse cells and add "human" in instances that refer to human cells. In the introduction, we included the following text:

"By contrast, converted mouse amniotic cells, or murine RACVECs (subsequently be referred to as simply, "RACVECs"), stably adopted an EC-like immunophenotype and acquired a transcriptome highly similar to cultured adult ECs."

Reviewers' comments:

Reviewer #1 (Remarks to the Author):

Info on genetic stability should be provided as soon as possible.

Reviewer #2 (Remarks to the Author):

I continue finding this story hard to follow and not particularly novel, since the roles of Sox17 and Erg1 are quite well established in endothelial fate determination. I see no particular novelty in the used of amniotic fluid cells.

Studies of CA-Akt1 are particularly unconvincing since, depending on the level and duration of expression, this can result in dominant-negative effects.

Reviewer #4 (Remarks to the Author):

The authors have answered to my previous comments and suggestions in a satisfactory manner. The manuscript is more improved and concrete.

Reviewer #1 (Remarks to the Author):

The authors responded adequately to my comments/suggestions. However their response P38 in regards to tumorigenicity is not sufficient. Confirmation of genetic stability/ normal karyotype should be performed. There are various approaches available.

Info on genetic stability should be provided as soon as possible.

We now include genomic sequencing results in **Supplementary Figure 1m** (shown below).

We have added the following text in the Results section:

We performed genomic sequencing of cultured MACs, RACVECs, and directly purified LSECs and, similar to human amniotic cells and human RACVECs, found minimal genetic alterations in the cultured cells compared to the directly purified adult ECs, indicating genomic stability (Supplementary Figure 1m).

We hope reviewer finds these data satisfactory and are happy to provide additional analyses and/or replicates as requested.

REVIEWERS' COMMENTS:

Reviewer #1 (Remarks to the Author):

the author responded adequately and I have no further suggestions/concerns.